# Using telemedicine to improve access, cost and quality of secondary care for people in prison in England: a hybrid type 2 implementation effectiveness study

Chantal Edge ,[1] Julie George ,[2] Georgia Black ,[1,3] Michelle Gallagher,[4] Aftab Ala,[4] Shamir Patel,[5] Simon Edwards,[6] Andrew Hayward[1,7]

For numbered affiliations see end of article.

**Correspondence to**
Dr Chantal Edge;
c.edge@ucl.ac.uk

## ABSTRACT

**Introduction** People in prison tend to experience poorer health, access to healthcare services and health outcomes than the general population. Use of video consultations (telemedicine) has been proven effective at improving the access, cost and quality of secondary care for prisoners in the USA and Australia. Implementation and use in English prison settings has been limited to date despite political drivers for change. We plan to research the implementation of a new prison-hospital telemedicine model in an English county to understand what factors drive or hinder implementation and whether the model can improve healthcare outcomes as demonstrated in other contextual settings.

**Methods and analysis** We will undertake a hybrid type 2 implementation effectiveness study to gather evidence on both clinical and implementation outcomes. Data collection will be guided by the theoretical constructs of Normalisation Process Theory. We will prospectively collect data through: (1) prisoner/patient focus groups, interviews and questionnaires, (2) prison healthcare, hospital and wider prison staff interviews and questionnaires, (3) routine quality improvement and service evaluation data. Up to four prisons and three hospital settings in Surrey (England) will be included in the telemedicine research, dependent on their telemedicine readiness during the study period. Prisons proposed include male and female prisoners, remand (not yet sentenced) and sentenced individuals and different security categorisations. In addition, focus groups in five telemedicine naïve prisons will provide information on patient preconceptions and concerns surrounding telemedicine.

**Ethics and dissemination** This study has received National Health Service Research Ethics Committee, Her Majesty's Prison and Probation Service National Research Committee and Health Research Authority approval. Dissemination of results will take place through peer-reviewed journals, conferences and existing health and justice networks.

## Strengths and limitations of this study

► This research is studying the real-time implementation of a complex digital intervention within prison settings.
► Peer researchers from the Prison Reform Trust are helping to deliver Patient Public Involvement and qualitative data collection in prisons, likely to improve patient disclosure and research engagement.
► This telemedicine intervention is not planned for delivery in any high-security (category A) prisons; therefore those who may experience the most problems accessing hospitals are not included in the patient post-telemedicine sample.

the community, despite numerous national and international directives which cite the right of prisoners to equivalence of healthcare.[1] [2] Access to healthcare in prisons must be operationalised within the security constraints of the prison environment. All prisons in England have primary care services on site; however, secondary care services will normally be accessed off-site at local hospitals, with prisoners escorted to these premises by prison officers, at cost to the National Health Service (NHS).[3] Given the resource requirement for off-site transfer and the high burden of disease experienced by people in prison, it is not unusual for patients to experience lengthy waits for non-urgent care before commencing treatment. If the need for an emergency appointment arises on the day, this will likely take precedence over the scheduled outpatient transfer. Patients may also be reluctant to attend off-site appointments due to the stigma they anticipate and experience in handcuffs at local hospital sites.[4] In addition, due to concerns that prisoners may make plans to abscond, prisoners are not permitted

## INTRODUCTION

Prisoners tend to have poorer access to secondary healthcare than people living in

to know the time and date of their off-site appointment, meaning they wait for an undetermined period of time with anxiety after referral.

One approach which has been used to address these issues is telemedicine. Telemedicine refers to the use of technology, including video link, to allow remote consultation for patients without the need for direct physical contact with local health services. Telemedicine consultations have been used in prisons worldwide to reduce inequities in healthcare access experienced by prisoners,[5–11] most notably in the USA and Australia.[12] There is a large body of evidence which demonstrates their effectiveness at improving the access and cost of healthcare provision in prisons.[13] The more limited data on patient experience suggests it also offers improved care quality for prisoners.[14–17]

The English prison system has not yet succeeded in implementing telemedicine at scale despite several previous attempts.[18–20] No current evidence exists thus far to describe why these models did not succeed and whether the lack of success was due to ineffective implementation or failure of the intervention itself. There is political impetus to introduce video consultations in England both in general secondary care sites as part of the NHS long-term plan digital agenda[21] and within prisons due to the anticipated improvements in both access to care and cost-effectiveness demonstrated by individual models elsewhere.[22–25]

Surrey is an English county (population 1.185 million)[26] housing approximately 2600 prisoners within 5 prisons, totalling 3% of the total prisoner population of England and Wales.[27] Surrey is commencing implementation of a local prison-hospital telemedicine model as a new way of delivering clinical care to prisoners. At the time of writing this protocol, community healthcare services (defined as healthcare provided outside prisons) are in a period of complex change as they seek to introduce new integrated healthcare delivery partnerships called Integrated Care Systems (ICSs).[28] The Surrey Heartlands ICS is looking to radically reduce face to face outpatient appointments with telemedicine likely to be one method deployed to meet this aim, and therefore prisons are acting as an early test bed for this intervention. Services deployed to prisons will focus on specific clinical areas at outset, namely hepatology, gastroenterology and sexual health. Sexual health services provided will initially focus on those provided by a health advisor role, for example, counselling regarding a sexual health diagnosis or contact tracing. Hepatology and gastroenterology consultations will include routine follow ups and treatment of hepatitis C and B. The telemedicine intervention and the care pathways designed for use will remain the responsibility of the NHS as part of standard clinical care. Her Majesty's Prison and Probation Service (HMPPS) is supportive of telemedicine implementation provided staff comply with several conditions surrounding its usage including the presence of a prison healthcare staff member during a telemedicine appointment alongside the patient.

Within this research study, we will seek to understand whether telemedicine implementation between English prisons and hospitals is feasible and acceptable (to service providers, frontline staff and patients), the perceived barriers to overcome and whether it produces the range of improved outcomes (eg, access) hypothesised by providers. Through our extensive engagement with senior stakeholders in the health and justice system in England, this model of evidence will be used to understand whether wider implementation of prison-hospital telemedicine should be attempted in other English regions, and if so, which elements comprise an effective implementation strategy.

## METHODS AND ANALYSIS
### Study design
We will undertake a hybrid type 2 implementation-effectiveness study[29] to study the implementation of prison-hospital telemedicine in England. Hybrid type 2 studies have both a primary implementation outcome and a primary clinical outcome and aim to generate evidence of a proven clinically effective intervention (prison telemedicine) on patient outcomes (access to secondary care) within a new context (English prisons).

The primary implementation outcome is the acceptability and feasibility of prison–hospital telemedicine in this context and will be measured at different staff levels from different providers and for patients. The primary clinical effectiveness outcome is access to secondary care for people in prison in England, with secondary outcomes referring to the quality and cost of hospital care using this intervention.

### Setting
Up to four prisons and three hospital settings in Surrey (England) will be included in the research, dependent on their telemedicine readiness during the study period. It is not possible to determine how many consultations will take place during the study period as hospital appointment numbers for prisoners are traditionally lower than the need within this population, given the barriers to accessing hospital care from prisons. The four prisons proposed include male and female prisoners, remand (not yet sentenced) and sentenced individuals and different security categorisations.

### Participants
#### Staff
There are two types of clinical provider staff who will be sampled in this project.
1. Prison healthcare staff who act as a contracted service within the Surrey prisons but who are primarily employed by a central London NHS Foundation Trust. These staff will be responsible for referring patients to local community hospitals from prisons, local implementation and operation of the telemedicine system in prisons, attendance in telemedicine consultations with

patients and liaison with prison staff for arrangement of patient transportation.

2. Hospital staff employed within Surrey community hospitals who will be delivering secondary care to patients in prisons over the telemedicine link. There are two Surrey hospitals who may be sampled for this research dependant on telemedicine readiness. Preimplementation interviews will select staff from across all organisations. Throughout this protocol, we refer to these groups as: prison healthcare staff, hospital staff or all staff (prison and community hospital staff combined).

Wider prison staff from participating prisons will also take part as research participants. This will reflect their role in scheduling patient offsite transfers and escorting patients to the prison healthcare department for telemedicine appointment.

### Staff sampling

Participants for interviews prior to and during implementation will be selected using a combination of purposive and snowball sampling to generate a key informant sample. Purposive sampling will identify initial interviewees known to have been involved in telemedicine implementation (eg, clinicians, governance, IT). Snowball sampling will help identify other relevant interviewees unknown to the research team (eg, commissioning leads, strategic directors) from those interviewed first. Interviewees will be selected from across all departments/prisons participating and will comprise a mix of staff at macro (board level), meso (departmental lead/managers) and micro levels (frontline delivery of telemedicine). This approach will result in a key informant sample,[30] guided both by the research teams' knowledge of staff involved in telemedicine implementation and staff member's further knowledge on who is critical to support implementation locally. A maximum of 30 interviews is anticipated prior to telemedicine implementation, permitting a wide understanding of how telemedicine is understood and supported across varying staff levels and provider organisations. Mapping of staff at macro, meso and micro levels across hospital and prison healthcare organisations has informed this sample size.[31]

At the end of the telemedicine data collection period, semistructured 1:1 interviews will be conducted with key staff informants from all clinical staff and wider prison staff (eg, prison officers). Interviews will be split among (a) staff who have experience of using/contributing to others use of telemedicine and (b) staff from wider specialties who may wish to use telemedicine for prison consultations in the future. Purposive and snowball sampling will be used to identify staff interview participants with a strategic view of the health system who can comment on wider clinical specialties that may be appropriate for delivery via telemedicine in prisons, based on lessons learnt from implementation of the current services.

Prison staff (eg, prison officers) from participating prisons will also be approached for participation in research activities, with participant numbers guided by the responsible prison governor.

### Prisoners

We will include prisoners from both telemedicine and telemedicine-naïve prisons as participants within this study.

1. Patients in telemedicine prisons who use the system for an appointment will be invited by prison healthcare staff to complete a postappointment questionnaire and an opportunity to participate in a 1:1 interview with the research team.
2. In conjunction with the Prison Reform Trust charity (PRT)[32] we will organise and run focus groups (n=5) in five 'telemedicine-naïve' prisons outside of the Surrey footprint (ie, those who have no prior experience of telemedicine/are not preparing for imminent telemedicine implementation). Participants will be recruited via existing PRT networks within the prisons.

## Consent

Informed consent for participation from healthcare staff will be taken by the researcher. Informed consent from prisoners in focus groups will be taken by the PRT charity. Informed consent from prisoners in interviews will be taken by the researcher in the presence of a prison healthcare staff member.

## Key inclusion and exclusion criteria
### Staff

For stage 1 staff interviews, staff must have a stakeholder role in local telemedicine implementation and delivery. For stage 2 acceptability/feasibility interviews stuff must have either been involved in implementing telemedicine intervention in Surrey or have used the intervention (either accompanying patients in consultations or providing secondary care treatment) or work as a staff member in a prison offering telemedicine. For stage 2, interviews on wider telemedicine applicability staff must have either responsibility for prioritising or approving off-site appointments to hospitals for patients in prisons or provide secondary care to offenders or involved in healthcare delivery to prisons including strategic level staff.

### Prisoners

Prisoner participation in any research activity in a prison is ultimately vetted and granted by prison staff in light of safety concerns regarding the individual. Several general exclusion criteria exist for participants in prisons which are common across all research activities proposed. These include: people who cannot give informed consent, people who are deemed a risk to researchers/ not permitted to participate (by prison officials). In addition patients who move prison after their telemedicine consultation will not be approached for interview unless they have moved to another one of the study prisons where HMPPS ethical approval is valid.

To participate in patient questionnaires or patient interviews, the prisoner must have used or been offered

use of the local telemedicine system. Participation in telemedicine focus groups will be open to all prisoners who are not excluded based on the criteria above and who currently reside in a telemedicine-naïve prison.

## Theoretical framework

Use of theory in implementation research both guides the collection and analysis of data and provides explanations for the phenomena observed.[33] Our implementation research will be informed primarily by Normalisation Process Theory (NPT)[34] which has been used to describe and explain implementation outcomes in numerous studies of complex healthcare interventions.[33]

NPT focuses primarily on the work that individuals and groups undertake to operationalise and normalise an intervention. The theory proposes four constructs: coherence, cognitive participation, collective action and reflexive monitoring.[34 35] As identified in a systematic review of prison telemedicine,[12] staff attitude and behaviour is reported to be both the main driver and barrier to successful implementation, making NPT particularly relevant to this intervention. Successful implementation elsewhere has found that staff at macro (board level), meso (departmental lead/managers) and micro levels (frontline delivery of telemedicine) must be engaged and enthused in the provision of the intervention, across a multiplicity of providers (community hospital, prison healthcare, wider prison staff) all who have differing needs, wants and beliefs.[12]

Most proposed evaluation activities will focus on staff perceptions of feasibility and acceptability. Data collection (eg, interview topic guides) will be guided by the constructs of NPT and research findings will be reported using a mixed methods approach under NPT construct headings.

NPT calls for a description of the intervention and the context in which it will be deployed. Through interviews and use of the NOrmalisation MeAsure Development (NOMAD) survey[35] we will gather staff data on these factors. NPT has previously been criticised for its focus on individual and collective agency and not paying enough attention to the wider organisational and relational contexts of the implementation.[33] Therefore, to provide more generalisable contextual information, in parallel to NPT, several constructs from the Consolidated Framework for Implementation Research (CFIR) will be used to guide an in-depth description of the intervention itself (core components and adaptable peripheries), a description of the outer setting (economic, political and social context) and the inner setting (structural, political and cultural contexts through which the implementation process will proceed).[36] In this way, CFIR will be used to identify potential local contextual implementation barriers and facilitators that will be specifically explored within the individual/collective domains of NPT, which in itself will provide explanation as to why these matter in this context and the interrelations among constructs.

NPT refers to the opinions and reflections of staff involved in the intervention under study and the work they undertake to deliver the intervention. In the general community, patients are also required to undertake 'work' to access healthcare appointments. For example, booking the appointment, attending the appointment and remembering to take their medication. However, given the limited autonomy of patients in prison their ability to perform this 'work' is restricted. This means that NPT does not necessarily form the best framework for understanding prisoner data, despite the fact that their views on telemedicine will indicate whether it has a chance of normalisation (ie, use). Prisoners will always be able to opt for standard care as usual instead of telemedicine. For this reason, prisoner data will be analysed using the Theoretical Framework of Acceptability (TFA)[37] which can be used to access prospective and retrospective acceptability of interventions to recipients where autonomy may be constrained. TFA constructs include: affective attitude, burden, ethicality, intervention coherence, opportunity costs, perceived effectiveness and self-efficacy. These will be used to guide both preimplementation and postimplementation data collection with patients.

## Patient and public involvement

Two PPI representatives with lived experience of imprisonment assisted with preparation of study materials prior to ethical submissions and attend steering group meetings in the community. Following National Research Committee (NRC) approval, PPI groups are being convened within prisons (one male, one female) to input into ongoing research activities for example, patient questionnaire design. These groups will each meet three times during the study, with continuity of membership if possible within the prison environment.

## Research activities and quality improvement measures

Research activities planned are shown in figure 1. Several quality improvement measures will run in parallel to research activities and complement the final analysis.

# PREIMPLEMENTATION RESEARCH ACTIVITIES
## Research activity 1 (RA1): staff interviews (stage 1)

1:1 Semistructured interviews (maximum n=30) among telemedicine stakeholders from all staff groups to understand anticipated benefits, barriers and enablers to telemedicine usage in a prison-hospital setting, within the English commissioning context. The interview guide has been developed based on the NPT constructs (see online supplementary file 1). These constructs provide a framework with which to understand the sense making of the intervention (telemedicine) among staff groups, the work they perceive will be involved in delivering the intervention and the benefits potentially accrued to system partners. Interview questions are aligned to those posed within the NOMAD questionnaire (see RA2), a survey measure

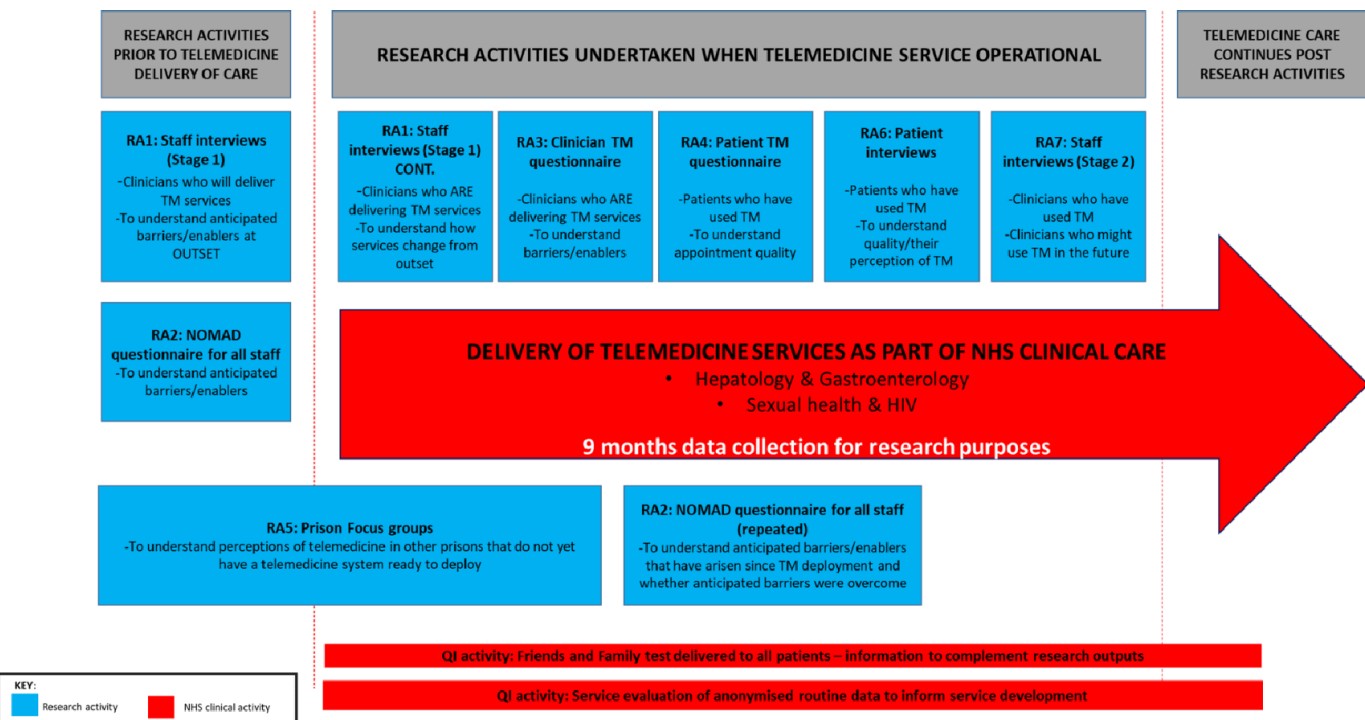

**Figure 1** Study research activities. NHS, National Health Service; RA, research activity; TM, telemedicine.

of NPT, but allow participants to expand widely on their answers to provide rich qualitative understanding.

## RA2: NOMAD questionnaire

The NOMAD questionnaire is an existing validated questionnaire, based on the principles of normalisation process theory (Finch, T.L., Girling, M., May, C.R. *et al.* Improving the normalization of complex interventions: part 2 - validation of the NoMAD instrument for assessing implementation work based on normalization process theory (NPT). *BMC Med Res Methodol* 18, 135 (2018). https://doi.org/10.1186/s12874-018-0591-x). The NOMAD questionnaire seeks to identify barriers to normalisation of an intervention in healthcare to explain why interventions fail to embed in day to-day use. We will administer this anonymous email survey questionnaire to all healthcare staff and additionally to prison officers at participating prisons. This will collect data on staff perceptions and concerns surrounding telemedicine. This questionnaire will be administered prior to telemedicine implementation and several months after implementation, and the changes in response quantified.

## RA5: prison focus groups

Study prisons have been advocating for telemedicine introduction for several years, and prisoners are aware it may soon be introduced. We will collect data to represent broader prisoner views on telemedicine in prison focus groups (n=5) where there is currently no 'threat' of implementation and prisoners will be free to speak openly of their concerns. Results will be used where applicable to inform the current telemedicine service implementation and to inform policy for future telemedicine

services if research results suggest it can be widely implemented. For example: "What do you think would stop people wanting to use a telemedicine system and how could we mitigate this?" The topic guide for focus groups has been guided by the TFA and PPI groups (see online supplementary file 2). The first topic guide iteration was prepared by the researcher, by applying TFA constructs (affective attitude, burden, ethicality, intervention coherence, opportunity costs, perceived effectiveness and self-efficacy) to the perceived telemedicine model in prison. For example, an opportunity cost to attending a telemedicine appointment may be missing a family visit or the opportunity to travel outside the prison walls. This topic guide was reviewed at two PPI groups (one male, one female) and refined in line with feedback.

## IMPLEMENTATION RESEARCH ACTIVITIES
These activities will commence after telemedicine has become operational in at least one prison site.

### RA1: staff interviews (stage 1) continued
Frontline delivery staff will be reinterviewed during implementation to understand how the telemedicine services proposed at outset have changed and adapted as issues arise and what specific barriers have arisen that were not anticipated. Staff will be approached for reinterview when the system has been running for a minimum of 4 months. They will be reinterviewed once only as part of RA1; however, may also be interviewed during RA7 (staff interviews at end of data collection period). Only staff involved in frontline service delivery will be reinterviewed

- Were you able to connect/log on to the system with ease?
- Was the quality of the internet connection sufficient for consultation?
- Was the telemedicine consultation sufficient to understand/treat patient clinical issues (or will an in-person appointment be required)?
- Were you able to discuss the onward treatment plan for the patient with the hospital/prison staff member?
- Would there have been difficulties in planning/delivering the patient treatment plan if you had not spoken directly with the hospital/prison clinical staff?
- Did the patient need your help to clarify any clinical information during or after the appointment?
- Did the patient appear comfortable in the telemedicine appointment?
- In hindsight, do you think the patient was appropriately referred to a telemedicine appointment? (and if not, why not?)

**Figure 2** Anticipated staff questions for questionnaire post-telemedicine consultation.

for RA1, and we anticipate a maximum of 10 interviews based on knowledge of staff numbers involved in initial telemedicine delivery.

### RA3: clinician questionnaire on telemedicine appointment

Both hospital clinicians and prison healthcare staff involved in a telemedicine appointment will complete a paper questionnaire at the close of each appointment. This will seek to understand whether the appointment was clinically acceptable, with reference to anticipated barriers identified in the staff interviews at outset. The content of the questionnaire will include questions from the Telemedicine Usability Questionnaire (TUQ)[38] which relate to staff actions for use of telemedicine. For example, dialling in to the telemedicine system will be the responsibility of the prison healthcare staff member and not the patient. Questions will also be informed by the literature review that precedes this protocol on enablers/barriers to prison telemedicine,[12] for example, "where you able to access prison electronic health records contemporaneously?".

The questionnaire will be refined after clinical staff interviews have been conducted to include questions around staff concerns identified. We currently anticipate the questions shown in figure 2.

### RA4: patient questionnaire on telemedicine appointment

All patients who have a telemedicine appointment will be offered a short, anonymous paper questionnaire immediately after their appointment finishes to collect data on patient acceptability of telemedicine appointments. Questionnaires should provide information that may be overlooked as 'not important' or not recalled in later patient interviews, for example, "Did the doctor speak over you?".

The content of the questionnaire will be based on the TUQ[38] with several additional questions pertaining to appointments in prison. For example, "Were you able to get to your appointment on time?", which refers to the fact that prison officers will need to escort patients to their telemedicine appointment within the prison healthcare department.

The questionnaire will be finalised with input from PPI groups. For example, recent PPI work has suggested questions including, "Have you used a video court link before?" and "Did the healthcare staff member dial in before you were in the room?" are likely to be particularly relevant to patient experience of prison telemedicine.

### RA6: patient 1:1 interviews

1:1 Semi-structured interviews (n=15) will be held with patients who have had a telemedicine consultation to collect information on patient experience and acceptability. Where applicable participants will be asked to compare telemedicine treatment to previous offsite secondary care appointments. It is not possible to determine in advance how many patients will have previous offsite secondary care experience given the dynamic nature of the prison population.

Patients who are offered the opportunity for a telemedicine appointment but do not choose to undertake it will be approached for interview to understand their views of telemedicine.

### RA7: staff interviews (stage 2)

Interviews with staff who have used the telemedicine system with seek to understand the feasibility and acceptability of the current delivery model. Interview topic guides will be informed by NPT constructs and results of the NOMAD survey previously administered.

All frontline healthcare staff who have used the model will be invited to participate in interview, as well as staff involved in operational or managerial support of the model (n=10–15). It is possible that prison healthcare teams may choose to assign only one or two individuals to chaperone telemedicine consultations during the mobilisation phase to ensure consistency, which may limit the interview sample size.

### Qualitative analysis

All qualitative data collection activities will be recorded on an encrypted digital voice recorder and professionally transcribed.

Focus group and interview data will be analysed using framework analysis, by which data is sifted, charted and sorted in accordance with key issues and themes. Framework analysis involves a five step process: (1) familiarisation; (2) identifying a thematic framework; (3) indexing; (4) charting and (5) mapping and interpretation.[39]

### Quality improvement measures
#### The NHS friends and family test

Alongside the research activities a standard NHS quality improvement tool (the friends and family test) will be administered by the prison healthcare staff to patients to monitor satisfaction of telemedicine services for all appointments. Anonymised data from this will be provided to the research team to supplement the research summary and will be compared with pretelemedicine satisfaction scores.

## Quantitative service evaluation—pre-anonymised and post-anonymised health records data for specialties under study

After 9 months of service delivery routine, service evaluation data will be used to assess clinical effectiveness, specifically to understand whether appointments happened more quickly using telemedicine and whether cancellations were reduced. These data will be collected and analysed by the NHS as part of their routine clinical evaluation of a service change. Aggregate anonymised electronic health records data will be provided to the research team to supplement the research summary including: number of secondary care consultations and referrals, time from referral to treatment, total number of off-site transfers, number of and reason/s for appointment cancellation. These data will be compared at stages throughout implementation with pretelemedicine data on the same indicators to account for a transition phase as the service becomes established.

## Quantitative analysis

We will seek to understand how the use of telemedicine changes the referral to treatment time for patients and the number of appointment cancellations. We will also compare this to equivalent indicators for patients from the general community to understand whether telemedicine can improve equity between these groups.

We will collect information currently lacking such as the distributional form and SD of key indicators of healthcare use such as referral to treatment time. This information can then be used in future research to calculate sample size and power as well as determining the most appropriate statistical tests to use in analysing the data. From the basic information on the number of cases, we might reasonably expect to use telemedicine for gastroenterology within one prison over the period of the study (n=50–55), we can state that if the intervention (telemedicine) has a medium to large effect size, the feasibility study will be powered to detect this difference, even with the relatively small sample size. Specifically, we estimate a value of 0.6 for Cohen's D which he equates to medium to large effect size.[40] These data will provide evidence as to whether telemedicine can provide a difference in care.

## Service evaluation to inform wider specialties for telemedicine delivery

To understand which other specialties may be a priority for prison telemedicine delivery aggregate, anonymised prison healthcare records data will be provided to the research team. This will provide information on the need and distribution of all secondary care outpatient appointments for people in participating prisons including: hospital and specialty referred to for example, cardiology, time between referral and appointment by specialty, cost of prison officer escorts for appointment and cancellation data (if applicable), that is, reason for cancellation and date of cancellation by specialty.

Indicators will be compared as proportions to allow for differences in prison population sizes and the dynamic population cohort. For example, number of cancelled appointments or number of referrals made to a hospital specialty will be presented per 100 standard general practitioner appointments within the prison.

This analysis will help inform which other hospital specialties may be areas of value for telemedicine development, for example, department with most referrals, department with longest wait, department with most cancellations. This will complement stage 2 staff interviews on wider telemedicine specialties.

Use of data collection tools throughout this research is detailed further in table 1.

## Economic analysis

Cost modelling will be undertaken by a health economist to understand how the wider roll out of prison telemedicine across specialties may affect cost-effectiveness of secondary care delivery to prisoners (including costs to prison healthcare provider, wider prison costs and NHS costs). Data on resource associated with delivery of telemedicine (eg, staff time and processes) will be collected in staff feasibility/acceptability stage 2 interviews and on the staff questionnaires. For example, questionnaires will collect data on "How long did you take to log on to the telemedicine system?" and "How long was the appointment you chaperoned?". Interviews will collect data such as "which staff grades are you comfortable with designating as a patient chaperone" or "what tasks did you forgo to chaperone a telemedicine appointment?".

Individual prisons, the prison healthcare provider and community hospitals will provide costings data associated with the delivery of the telemedicine model compared with standard care which will be attributed to reported data.

## Confidentiality

Qualitative data will be anonymised in published reports and on archiving at the close of the study to protect participant confidentiality. Interview/focus group data will be recorded on an encrypted dictaphone. Service evaluation data will be provided as aggregate, anonymised data to remove the risk of deductive disclosure. Questionnaires from both staff and patients will be anonymous.

## Ethics and dissemination

Study results will be published in peer-reviewed journals and presented at academic conferences. Wider dissemination will be achieved through health and justice networks/organisations such as Public Health England, The Prison Reform Trust and the Royal College of General Practitioners Secure Environment Group.

Ongoing feedback will be provided to prison Governors by the lead researcher, both verbally and by email communication. The Prison Reform Trust will ensure research summaries are made available that are accessible to people in prison.

**Table 1** Data collection instruments

| Data collection instrument | Administered to | Purpose | Point of administration | Administered by | Analysis plan |
|---|---|---|---|---|---|
| NOMAD (NPT) survey | Staff involved in operation, delivery or support of prison telemedicine (hospital staff, prison healthcare staff, prison officers). | To understand staff preconceptions and concerns that may affect implementation and normalisation of telemedicine. This wider survey will complement in-depth staff interviews. | Prior to telemedicine implementation and repeated at end of telemedicine data collection period. | Online survey sent through email link. Link circulated to staff by provider/ prison leads (eg, local principle investigators). | Descriptive analysis and comparison of survey data prior to/post implementation. Descriptive analysis will be triangulated with qualitative data on NPT constructs (from staff interviews) to understand how telemedicine is perceived and understood by staff. |
| Staff telemedicine questionnaire (informed by the TUQ) | Staff either delivering or chaperoning prison telemedicine consultations (hospital staff, prison healthcare staff). | To understand the clinical acceptability and feasibility of telemedicine appointments to staff, limitations and staff time associated costs of telemedicine. This questionnaire will complement in-depth staff interviews. | To be completed by the staff member at the end of each telemedicine consultation. | Paper questionnaire for self-completion available at study sites. | Descriptive analysis of survey measures. |
| Patient telemedicine questionnaire (informed by the TUQ) | Patients who have had a telemedicine appointment. | To understand the acceptability of telemedicine appointments to patients. This questionnaire will complement in-depth patient interviews. | To be completed by the patient after a telemedicine consultation (can be taken away for completion). | Paper questionnaire for self-completion available at study sites. Prison staff or peer healthcare representatives will assist with completion if required for literacy reasons. | Descriptive analysis of survey measures and qualitative analysis of free text answers in NVIVO software. Qualitative data from surveys will be analysed alongside interview data pertaining to patient experience following the principles of framework analysis. |
| NHS Friends and Family Test | All patients at the prison complete this tool as part of usual care. Survey data will be collated for prison telemedicine appointment subsets. | To understand whether patient satisfaction of healthcare improves with telemedicine. | To be completed by the patient after a telemedicine consultation. | Paper questionnaire for self-completion available at study sites. | Descriptive analysis and comparison to overall historic departmental trends in healthcare satisfaction. |

NHS, National Health Service; TUQ, Telemedicine Usability Questionnaire.

## DISCUSSION

The potential for the prison context to restrict research activities should not be underestimated. Undertaking research in prisons is complex given the restrictive nature of the environment and the need to operate within the prison regime, which will inevitably differ by establishment. Health research approvals must be granted by the research sponsor, an NHS research ethics committee, the Health Research Authority (who provide general health service agreement) and individual NHS hospitals. In addition, for research taking place in prisons, the HMPPS National Research Committee must provide approval which, alongside methodological scrutiny, they grant in line with their guidance:

> All research must be of benefit to HMPPS and the links to HMPPS priorities must be explicit.[41]

Finally, individual prison governors must provide agreement that research can take place within their given establishment.

Data collection is also more challenging than in community settings. Prison electronic health records currently sit separately from community NHS records and are jointly controlled by both prison healthcare provider and NHS prison commissioners. Data collection direct from prisoners is reliant on the ability to access patients. When using questionnaires, these must account for a population which is likely to have low literacy levels and an element of mistrust in research. In addition, as identified in our forthcoming literature review,[12] prison and hospital healthcare providers are likely to anticipate and experience different and unequal benefits, barriers and facilitators which may affect enthusiasm for implementation.

To undertake this research, the principal investigator (PI) is assuming the role of an embedded researcher, which is known to present challenges to rigorous research within healthcare organisations.[42] In this instance, the PI will be embedded both within the community healthcare team/s and the prison healthcare team alongside their academic base. This may compromise their ability to truly embrace 'dual' affiliation and relationship building given that relationships must be formed with two different hospital provider teams.

In summary, a telemedicine model straddling community and prison healthcare providers, situated within the context of the English prison system represents a highly complex intervention. For this reason, we have determined it appropriate to only study prisons within one county of England, which houses both male and female prisons of different security categorisation. This may restrict our sample size; however, attempts to widen the study to other counties would introduce other prisons, other hospitals, other providers and other commissioners and leave the study prone to failure as we strive to understand whether successful implementation is possible.

## Potential significance of research

If established effectively, prison-hospital telemedicine offers the potential to improve access and quality of secondary care for prisoners and to reduce NHS costs by reducing the amount of offsite escorts from prison to hospital that must be reimbursed from public NHS funds. There is an opportunity cost to all NHS spending, with one funded service or treatment representing an opportunity foregone to invest elsewhere.

Prisoners experience a disproportionately higher burden of disease than people within the general community including infectious diseases, long-term conditions and mental health problems.[43] The prison population is typically characterised as younger, but the number of older prisoners continues to rise, bringing new challenges in the form of treatment of multiple comorbidities within prison environments.[44] Telemedicine may be one approach to improve the health of people in prisons, which can in turn bring community dividends such as

reducing reoffending and a reduction in the wider reservoir of infectious disease.[45 46] However, it cannot hope to bring these benefits for a traditionally underserved and under-researched population, unless the implementation is successful.

In addition, the study of implementation science within criminal justice settings is highly limited. At the time of writing this protocol, we have identified only three articles pertaining to implementation research in health and justice settings.[47–49] This study will therefore contribute knowledge to an emergent and complex field of study.

**Author affiliations**

[1]Collaborative Centre for Inclusion Health, University College London, London, UK
[2]Institute of Health Informatics, University College London, London, UK
[3]Department of Applied Health Research, University College London, London, UK
[4]Gastroenterology and Hepatology, Royal Surrey County Hospital NHS Foundation Trust, Guildford, UK
[5]Offender Care, Central and North West London NHS Foundation Trust, London, UK
[6]Diggory Division, Central and North West London NHS Foundation Trust, London, UK
[7]Institute of Epidemiology and Health Care, University College London, London, UK

**Acknowledgements** We are grateful to the University College London Joint Research Office and the Research and Development departments at Royal Surrey Hospital NHS Foundation Trust and Central North West London NHS Foundation Trust for their support and assistance in developing this research study.

**Contributors** CE conceived the project. CE wrote the original research grant funding application and protocol, with methodological guidance and support from academic supervisors AH, JG and GB. SE, SP, MG, AA are senior clinicians within participating hospitals who have guided methodological considerations throughout and provide clinical supervision of the PhD student. SP and MG are named local principal investigators, AH is named Chief Investigator for the study for ethical approvals. CE is named Principal Investigator and PhD student. All authors reviewed and commented on the final draft of this protocol.

**Funding** CE is funded by a National Institute for Health Research (NIHR), (Clinical Doctoral Research Fellowship ICA-CDRF-2017-03-006) for this research project. JG is partially funded by a Health Education England/National Institute of Health Research Clinical Lectureship (ICA-CL-2016-02-024). AH is an NIHR senior investigator. This publication presents independent research funded by the National Institute for Health Research (NIHR). Study Sponsor: University College London (Joint Research office) email: uclh.jro-communications@nhs.net.

**Competing interests** None declared.

**Patient consent for publication** Not required.

**Ethics approval** This study has received ethical approval from the London South East NHS Research Ethics Committee (19/LO/0098), the HMPPS National Research Committee (NRC 2018-212) and Health Research Authority and Health and Care Research Wales (HRA IRAS 229646).

**Provenance and peer review** Not commissioned; externally peer reviewed.

**Author note** CE is undertaking this study as a PhD study which forms the basis of her NIHR Clinical Doctoral Research Fellowship. The named chief investigator for the study is her primary PhD supervisor AH in line with University College London study sponsorship policy.

**ORCID iDs**
Chantal Edge http://orcid.org/0000-0002-1439-0826
Julie George http://orcid.org/0000-0003-2410-2696
Georgia Black http://orcid.org/0000-0003-2676-5071

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
