## [Reviewer comments · BMJ Open]

ARTICLE DETAILS

TITLE (PROVISIONAL)	Using Telemedicine to Improve Access, Cost, And Quality of Secondary Care for People in Prison in England: A Hybrid Type 2 Implementation Effectiveness Study
AUTHORS	Edge, Chantal; George, Julie; Black, Georgia; Gallagher, Michelle; Ala, Aftab; Patel, Shamir; Edwards, Simon; Hayward, Andrew

VERSION 1 – REVIEW

REVIEWER	Sandra Thompson University of Western Australia, Australia
REVIEW RETURNED	15-Dec-2019

GENERAL COMMENTS	This study protocol is clear and well written, including contextualising the challenges with undertaking such a study in the prison setting and why it is restricted to one county. This is an important study and the authors are to be congratulated for their work in developing the study. It will provide important information. My specific comments are: The specific clinical areas of sexual health, hepatology and gastroenterology were selected. It might be useful to elaborate on what conditions this is likely to include based upon existing need for consultations - I imagine this includes treatment of hepatitis B, hepatitis C and HIV but no doubt there are other conditions as well. The protocol refers to analysis but does not generally specify how quantitative data will be analysed, statistical testing. The authors state that it will be difficult to predict the number of telehealth consultations. They should provide some indication of numbers of interviews – for example in RA1 and RA7 and their sampling approach (I note it is purposive and snowballing). What will determine the numbers? The statement (RA1) that frontline staff will be reinterviewed during implementation is slightly ambiguous – when will they be re-interviewed and could this be multiple times? Given the importance of the prison governors for this research, it may be useful to consider approaches to share the progress and knowledge gained during the study with governors –some information regarding this could be added. Given the importance of the economic analysis, I felt the specifics of data collection and analysis in this section is underdone. It
--

	reads as if this is to be outsourced to a health economist but more specific information on this would be useful. The current statement says data will come from staff feasibility /acceptability interviews but relevant questions for this do not appear in Figure 2. There are a number of different instruments which will be used for data collection e.g. NOMAD, TUQ, Friends and Family test. I think it would be useful for the reader to have a table which lists each instrument, perhaps includes a summary of its purpose, who will administer the instrument, when, and how data will be analysed.
--	---

REVIEWER	David Shaw, Bernice Elger IBMB University of Basel, Switzerland
REVIEW RETURNED	19-Dec-2019

GENERAL COMMENTS	This is a well-described protocol of an important study on implementing telemedicine in prisons in one English county. It is generally of a very high standard, and the design seems appropriate to the aims. However, I have a few concerns about biased recruitment potentially affecting the results. In telemedicine prisons, focus is on those who use the service; prisoners who do not use the service will not be asked, even if they have prior hospital visit experience. Under current design no-one from telemedicine prisons who has not used the service will be included in one-to-one interviews or in focus groups. Researchers state: "We will collect data to represent broader prisoner views on telemedicine in prison focus groups (n=5) where there is currently no 'threat' of implementation and prisoners will be free to speak openly of their concerns" but these are only in naive prisons. What about the views of prisoners who don't want to use telemedicine at all but are frequent hospital attenders in prisons with telemedicine? Could the results not be skewed by omitting this group? As prisoners in prisons with telemedicine they might have more informed negative views of it than those in naive prisons, even if they haven't used the service. Finally, what proportion of those who use telemedicine will have had hospital experience as prisoners to compare/contrast? Richest data would come from those who have used both.
---

VERSION 1 – AUTHOR RESPONSE

Reviewer: 1

I'd like to thank the reviewer for such positive feedback on this study.

The specific clinical areas of sexual health, hepatology and gastroenterology were selected. It might be useful to elaborate on what conditions this is likely to include based upon existing need for consultations - I imagine this includes treatment of hepatitis B, hepatitis C and HIV but no doubt there are other conditions as well.

- I have included some further information on consultations that will be delivered initially. As part of the clinical operation the Consultant involved will individually triage which consultations they believe they

can deliver remotely from all referrals made. Understanding their perception of clinical acceptability will form part of the final set of staff interviews. A more specific list of conditions treated will emerge as part of the implementation.

The protocol refers to analysis but does not generally specify how quantitative data will be analysed, statistical testing.

- I have provided further information on quantitative analysis. Because this is an implementation study of a new intervention, we have relatively little information on which to base a sample size calculation. In the body of the paper, we demonstrate what size of difference we are likely to be able to demonstrate given the number of consultations we can reasonably expect. Provided we have sufficient numbers to report statistical differences we will undertake significance testing; otherwise we will simply report descriptive statistics

The authors state that it will be difficult to predict the number of telehealth consultations. They should provide some indication of numbers of interviews – for example in RA1 and RA7 and their sampling approach (I note it is purposive and snowballing). What will determine the numbers?

-We have provided a short description of how purposive and snowball sampling will be used. Further information on proposed sample sizes have also been given throughout the RA and introductory paragraph on sampling.

The statement (RA1) that frontline staff will be reinterviewed during implementation is slightly ambiguous – when will they be re-interviewed and could this be multiple times?

-This has been clarified in the protocol.

Given the importance of the prison governors for this research, it may be useful to consider approaches to share the progress and knowledge gained during the study with governors –some information regarding this could be added.

-A section on results dissemination has been added

Given the importance of the economic analysis, I felt the specifics of data collection and analysis in this section is underdone. It reads as if this is to be outsourced to a health economist but more specific information on this would be useful. The current statement says data will come from staff feasibility /acceptability interviews but relevant questions for this do not appear in Figure 2.

-I have provided further information on the data we hope to collect to inform this analysis.

There are a number of different instruments which will be used for data collection e.g. NOMAD, TUQ, Friends and Family test. I think it would be useful for the reader to have a table which lists each instrument, perhaps includes a summary of its purpose, who will administer the instrument, when, and how data will be analysed.

-This has been included as a new file – table 1.

Reviewer: 2

I'd like to thank the reviewer for their positive feedback and comments.

What about the views of prisoners who don't want to use telemedicine at all but are frequent hospital attenders in prisons with telemedicine? Could the results not be skewed by omitting this group? As prisoners in prisons with telemedicine they might have more informed negative views of it than those in naive prisons, even if they haven't used the service.

-Thank you to the reviewer for this important comment. This point was also raised in a PPI group since we submitted the protocol. We have therefore decided to include those who were offered the

opportunity of a telemedicine appointment but did not choose to take it in our patient interview sample. This change has been updated within the protocol accordingly.
 By default most of the telemedicine naïve focus groups include people who have experience of numerous hospital appointments, as the PRT ensure a broad selection of participants are recruited. With the above change in protocol we will be able to compare the views of prisoners who do not have access to telemedicine and those who have actively declined this opportunity.

Finally, what proportion of those who use telemedicine will have had hospital experience as prisoners to compare/contrast? Richest data would come from those who have used both.

- I agree with the reviewer and we will prioritise interviews with those who have experienced both types of appointment. At present unfortunately it is not possible to determine what this proportion will be as the prison population is highly dynamic and individual establishments can churn by numbers of greater than 100 in one week. Therefore we cannot be sure what the patient cohort will be during the data collection period, and what their prior experience may be. If few people are able to compare these experiences this will be noted as a limitation in reporting of results.

VERSION 2 – REVIEW

REVIEWER	Sandra Thompson Western Australian Centre for Rural Health, University of Western Australia Australia Australia
REVIEW RETURNED	03-Feb-2020

GENERAL COMMENTS	You have done a very comprehensive job on revising the manuscript and the extra detail strengthens the paper. There are a couple of minor errors in the manuscript and some missing punctuation which I am sure will be sorted out in proofing the article. e.g. page 9 . "where" instead of "were" ... prison telemedicine,(12) for example "where you able to access prison electronic health records contemporaneously?". page 12 Suggest remove the "he" in "Specifically, we estimate a value of 0.6 for Cohen's D which he equates to..." Congratulations on your design of an important and challenging study.
---